# *Lactobacillus delbrueckii* Protected Intestinal Integrity, Alleviated Intestinal Oxidative Damage, and Activated Toll-Like Receptor–Bruton’s Tyrosine Kinase–Nuclear Factor Erythroid 2-Related Factor 2 Pathway in Weaned Piglets Challenged with Lipopolysaccharide

**DOI:** 10.3390/antiox10030468

**Published:** 2021-03-16

**Authors:** Fengming Chen, Jiayi Chen, Qinghua Chen, Lingyuan Yang, Jie Yin, Yinghui Li, Xingguo Huang

**Affiliations:** 1College of Animal Science and Technology, Hunan Agricultural University, Changsha 410128, China; cfming@stu.hunau.edu.cn (F.C.); jiayi@stu.hunau.edu.cn (J.C.); chqh314@163.com (Q.C.); lingyuan_yang@yeah.net (L.Y.); yinjie2014@126.com (J.Y.); 2Hunan Co-Innovation Center of Animal Production Safety, CICAPS, Changsha 410128, China; 3The Engineering Research Center of Feed Safety and Efficient Utilization, Education Ministry, Hunan Agriculture University, Changsha 410128, China

**Keywords:** *Lactobacillus delbrueckii*, intestinal integrity, oxidative stress, TLR–Btk–Nrf2, weaned piglets, LPS

## Abstract

Oxidative stress is increasingly being recognized as a player in the pathogenesis of intestinal pathologies, and probiotics are becoming an attractive means of addressing it. The present study investigated the effects of dietary supplementation with *Lactobacillus delbrueckii* (LAB) on intestinal integrity and oxidative damage in lipopolysaccharide (LPS)-challenged piglets. A total of 36 crossbred weaned piglets (Duroc × Landrace × Large Yorkshire) were randomly divided into three groups: (1) non-challenged controls (CON), (2) LPS-challenged controls (LPS), and (3) 0.2% LAB (2.01 × 10^10^ CFU/g) + LPS treatment (LAB + LPS). On the 29th day of the experiment, the LPS and CON groups were injected intraperitoneally with LPS and saline at 100 ug/kg body weight, respectively. The results show that the LPS-induced elevation of the serum diamine oxidase (DAO) level and small intestinal crypt depth (CD) were reversed by the dietary addition of LAB, which also markedly increased the ileal expression of tight junction proteins (occludin, ZO-1, and claudin-1) in the LPS-challenged piglets. Furthermore, LAB supplementation normalized other LPS-induced changes, such as by decreasing malondialdehyde (MDA) in both the serum and intestinal mucosa and 8-hydroxy-2-deoxyguanosine (8-OHdG) in the jejunal mucosa, increasing glutathione reductase (GR) and glutathione peroxidase (GSH-Px) in both the serum and intestinal mucosa, and increasing glutathione (GSH) and superoxide dismutase (SOD) in the jejunal mucosa. LAB also activated Toll-like receptor (TLR)–Bruton’s tyrosine kinase (Btk)–nuclear factor erythroid 2-related factor 2(Nrf2) signaling pathways in the intestine, suggesting that it plays a vital role in the ameliorative antioxidant capacity of weaned piglets. In summary, LAB increased intestinal integrity by improving the intestinal structure and tight junctions while enhancing antioxidant functions via the activation of the TLR–Btk–Nrf2 signaling pathway.

## 1. Introduction

Weaning is an important developmental stage for newborn mammals. The early weaning of piglets maximizes sows’ reproductive performance and annual litter size, which are of great significance for the economic efficiency of pig production [1]. However, changes in the environment and feed can render newborn piglets highly susceptible to different stresses during weaning. Numerous studies have shown that weaning stress not only induces intestinal inflammation but also disturbs free-radical metabolism and antioxidant systems, resulting in severe oxidative stress [2,3]. Oxidative stress results from the disruption of the dynamic balance between the body’s production of reactive oxygen species (ROS) and antioxidant defense capacity [4]. When piglets undergo weaning stress, the expression of antioxidant enzymes such as glutathione peroxidase (GSH-Px) and superoxide dismutase (SOD) is downregulated in the intestinal mucosa, accompanied by an increase in malondialdehyde (MDA) content [5]. Weaned piglets tend to exhibit an enhanced intestinal permeability to endotoxins, leading to an increase in local or systemic inflammatory reactions [6,7]. Therefore, alleviating the negative effects of oxidative stress in weaned piglets is crucial for the development of the pig industry.

The proper application of probiotics, especially the resident microflora in the intestinal tracts of humans and most animal *Lactobacillus* species, reportedly improves the intestinal microecological environment, stabilizes the intestinal mucosal barrier, and prevents and alleviates intestinal diseases [8,9]. Furthermore, *Lactobacillus* can inhibit oxidative damage by scavenging ROS [10]. For instance, LAB can reduce ROS such as peroxide radicals, superoxide anions, and hydroxyl radicals [11]. Studies in pigs have shown that adding *Lactobacillus fermentum* to the diet increases serum SOD and GSH-Px, hepatic catalase (CAT), and muscle SOD [12]. However, the effects of Lactobacillus on intestinal redox homeostasis have not been fully elucidated.

Intestinal epithelial cells that are directly in contact with intestinal microbes have different pattern recognition receptors (PRRs), such as Toll-like receptors (TLRs) [13]. The PRR recognition of conserved microbe-associated molecular patterns results in signaling cascades. Nuclear factor erythroid 2-related factor 2 (Nrf2), a sensor for oxidative stress, mainly regulates the expression of genes that protect against oxidative-stress-mediated damage [14]. Many studies have shown that probiotics may affect the redox state of the body by modulating the Nrf2 signaling pathway, thereby maintaining homeostasis [15,16,17]. In addition to involve the Nrf2 signaling pathway, the gene expression of some antioxidant enzymes such as hemeoxygenase-1 (HO-1) can also be regulated through the TLR–Bruton’s tyrosine kinase (Btk) signaling pathway [18]. Our laboratory has demonstrated the beneficial role of *Lactobacillus delbrueckii* (LAB) in improving oxidative-stress symptoms in both suckling piglets and weaned piglets [19,20]. Based on these findings, we hypothesized that LAB could regulate redox homeostasis and barrier function in the intestine by activating the TLR–Btk–Nrf2 signaling pathway. The current study explored whether LAB administration could improve antioxidant capacity and intestinal barrier function in an oxidative-stress piglet model.

## 2. Materials and Methods

### 2.1. Bacterial Strain

The LAB CCTCC M 207,040 strain used in this study was provided by the microbiology laboratory of the College of Animal Science and Technology, Hunan Agricultural University. The potent preparation of LAB was added to the experimental diet at a dose of 2.01 × 10^10^ CFU/g, whose determination and addition were conducted as previously described [20].

### 2.2. Animals and Experimental Design

All the procedures were approved by the Animal Welfare Committee of Hunan Agricultural University, Changsha, China (ACC2019016). Thirty-six crossbred piglets ((Duroc × Landrace × Large Yorkshire), 7.14 ± 0.16 kg) weaned at the age of 25 d were randomly divided into three treatment groups, with six replicates and two piglets per replicate: (1) the control group (CON; a basal diet and injection of 0.9% NaCl solution); (2) the LPS group (LPS; a basal diet and injection of LPS at 100 ug/kg body weight); (3) the 0.2% LAB + LPS group (LAB + LPS; a basal diet supplemented with 0.2% LAB and an injection of LPS). The basal diet was based on corn–soybean meal (Table 1); and it was formulated as recommended by the National Research Council (NRC 2012) [21]. All the piglets had free access to food and water throughout the 28 d feeding trial. At eight o’clock in the morning on Day 29, the challenged groups were injected with LPS from *Escherichia coli* (*Escherichia coli* O55:B5, Sigma Chemical, St. Louis, MO, USA) at a dose of 100 ug/kg body weight (BW); the CON group was injected with saline. The dosage and injection of LPS were chosen based on previous studies with weaned piglets [22]. The initial and final BWs were measured individually, and food consumption was recorded daily.

### 2.3. Sample Collection

Four hours after the injection of LPS, six piglets in each treatment group (one per replicate) with body weights close to the average were killed by euthanasia with an i.v. injection of sodium pentobarbital (40 mg/kg body weight). Blood samples were collected in 10 mL tubes and centrifuged at 3000 g for 15 min at 4 °C, and the supernatants (serum) were collected and stored at −80 °C until subsequent analysis. Two 5-cm-long segments and one 10-cm-long segment were acquired from the middle jejunum and ileum. One 5-cm segment was collected in 4% paraformaldehyde in phosphatebuffered saline (PBS) for histologic analysis. The other 5-cm segments were collected for western-blotting analysis. The 10-cm intestinal segments were used for collecting mucosal samples. After harvesting, samples were immediately frozen in liquid nitrogen and stored at −80 °C until analysis.

### 2.4. Intestinal Morphology Analysis

Jejunal and ileal samples were collected and immediately fixed in 4% paraformaldehyde. After being sectioned (5 mm), the samples were stained with hematoxylin and eosin. The villous height and crypt depth were measured using the Leica DM3000 microscope (Leica, Wetzlar, Germany), and the villous height/crypt depth ratio (VCR) was calculated.

### 2.5. Measurement of Antioxidant Indices and Serum Diamine Oxidase Activity

Approximately 0.1 g of frozen mucosa was precisely weighed, homogenized in ice-cold saline at a ratio of 1:10, and centrifuged at 10,000× *g* for 10 min at 4 °C to collect the supernatants. The protein content of the supernatants was determined with the bicinchoninic acid (BCA) protein assay kit (P0010, Beyotime Biotechnology, Shanghai, China). The intestinal mucosal supernatants and serum were measured for MDA, glutathione (GSH), oxidized glutathione (GSSG), catalase (CAT), GSH-Px, glutathione reductase (GR), and SOD, using assay kits in accordance with the manufacturer’s instructions (Nanjing Jiancheng Bioengineering Institute, Nanjing, China). Diamine oxidase (DAO) activity in the serum and 8-hydroxy-2-deoxyguanosine (8-OHdG) were detected using ELISA kits according to the manufacturer’s instructions (Jiangsu Yutong Biotechnology Co., Ltd., Nanjing, China).

### 2.6. Protein Expression Analysis by Western Blotting

The jejunal and ileal tissue lysates were extracted and homogenized using a commercial lysis buffer. The protein concentrations were determined with an Enhanced BCA Protein Assay Kit (Beyotime Biotechnology, Shanghai, China). Equal amounts of protein were separated via SDS-PAGE and then blotted onto a membrane, which was incubated with a primary antibody overnight at 4 °C, and then with a secondary antibody conjugated to horseradish peroxidase (HRP) for 1 h at room temperature. The blots were developed with an enhanced chemiluminescence (ECL) kit (Thermo Scientific, Wilmington, NC, USA) and visualized using a Luminescent Image Analyzer ChemiDoc MP (Bio-Rad, Hercules, CA, USA). The antibodies used included anti-Zonula occluden-1 (ZO-1) (21773-1-AP), anti-Occludin (27260-1-AP), anti-Claudin-1 (13050-1-AP), anti-TLR2 (ab92946), anti-TLR4 (ab183459), anti-Btk (ab101233), anti-HO-1 (ab13248), anti-Nrf2 (ab92946), and anti-β-actin.

### 2.7. Statistical Analysis

The results are presented as the means ± standard errors of the means (SEMs). The statistical significance was analyzed via one-way analysis of variance (ANOVA), followed by Duncan’s multiple range test (SPSS 26.0 software, Chicago, IL, USA). All statements of statistical significance are based on probability values ≤ 0.05. All the figures were drawn using Prism 8.0.

## 3. Results

### 3.1. Growth Performance

No significant differences were observed in the final BW, average daily gain, average daily feed intake, and feed-to-gain ratio among the CON, LPS, LAB + LPS groups (*p* > 0.05) (Table 2).

### 3.2. Intestinal Morphology

As shown in Table 3, the LPS challenge significantly increased the CD in both the jejunum and ileum and decreased the VCR in the ileum (*p* < 0.05). However, LAB supplementation significantly reduced the CD in both the jejunum and ileum (*p* < 0.05).

### 3.3. Intestinal Permeability and Tight Junction Protein Expression

Serum DAO activity is a major biomarker of intestinal permeability. Figure 1A shows that the LPS-challenged group had higher (*p* < 0.05) serum DAO levels than the non-challenged groups, while LAB supplementation reduced the serum DAO levels in the LPS-challenged piglets (*p* < 0.05).

Figure 1B and C present the protein expression of the tight junction protein in the jejuna and ilea of the piglets. ZO-1 and claudin-1 expression in the ileum was reduced by LPS challenge compared to control but increased by LAB supplementation (*p* < 0.05). Moreover, LPS also reduced ZO-1 expression in the jejunum (*p* < 0.05). LAB supplementation increased the protein expression of occludin and claudin-1 in the jejunum, and occludin, ZO-1, and claudin-1 in the ileum (*p* < 0.05), but had no significant effect on ZO-1 expression in the jejunum in the LPS-challenged piglets (*p* > 0.05).

### 3.4. Intestinal Mucosal and Serum Antioxidative Indices

Table 4 show that LPS challenge markedly increased the serum MDA contents but decreased CAT, GSH-Px, GR, and SOD activity in serum (*p* < 0.05). Compared to the LPS group, the piglets receiving LAB had lower MDA contents but higher GR activity in serum (*p* < 0.05).

Compared with the CON group, challenge with LPS increased 8-OHdG and MDA contents; decreased CAT and GSH-Px activity, and GSH contents in jejunum (*p* < 0.05) (Figure 2). Relative to the LPS group, piglets in the LAB + LPS group had lower MDA contents but higher SOD activity in jejunum (*p* < 0.05).

As shown in Figure 2, LPS challenge markedly increased ileal MDA contents (*p* < 0.05). LPS-challenged piglets also showed lower GSH contents, and CAT, GSH-Px and GR activity in ileum (*p* < 0.05). Piglets receiving LAB administration had higher CAT and GR activity in ileum compared with the LPS group (*p* < 0.05).

### 3.5. Protein Expression of TLRs, Btk, and Nrf2

The protein expression levels of the TLRs, Btk, Nrf2 and their downstream genes are represented in Figure 3 and Figure 4. The expression of TLR2, TLR4, and Btk in the jejunum, and Btk, HO-1, and Nrf2 in the ileum significantly decreased (*p* < 0.05) in the LPS-group piglets. LAB supplementation markedly increased (*p* < 0.05) the expression of TLR2, TLR4, Btk, HO-1, and Nrf2 in the jejunum compared with that in the LPS-only group. The protein expression of ileal TLR4, Btk, and Nrf2 in the LPS-challenged piglets was also increased (*p* < 0.05) via LAB supplementation.

## 4. Discussion

LPS, Gram-negative bacterial outer membrane component, triggers the inflammatory response, leading to the release of large numbers of endogenous inflammatory mediators, including tumor necrosis factor (TNF-α), interleukins (IL-4, IL-10, IL-13, IL-1), chemokines, adhesion molecules, ROS, and reactive nitrogen species (RNS) [23,24]. In recent years, evidence has suggested that chemical processes involved in redox triggering oxidative stress play a key role in inflammatory pathophysiology. Oxidative stress may cause cell death and the breakdown of extracellular matrix (ECM), while necrotic cells and damaged ECM then release various intracellular and extracellular molecules, which act as ‘alarmins’ triggering inflammatory cascades through recognition by PRRs [25]. Since inflammatory response is inextricably linked to oxidative stress, the LPS-induced oxidative stress model is also widely accepted [22,26]. In this study, we used this model to investigate whether LAB supplementation could alleviate oxidative stress in piglets. Currently, many probiotics, especially *Lactobacilli*, are gradually being developed as alternatives to antibiotics and treatment strategies for post-weaning syndrome. These probiotics exert their effects through a variety of mechanisms, including stimulating the immune response, preventing the invasion of pathogens, and producing antibacterial substances [27]. Previously, our laboratory observed that average daily feed intake (ADG) and body weight at Day 21 were increased by the oral administration of LAB to preweaning piglets during the lactation stage [28]. Moreover, Lan et al. [29] and Liu et al. [30] reported that probiotic supplementation could improve the growth performance of piglets. However, in the current study, LAB administration had no effect on the growth of piglets, which is consistent with the results of Ross et al. [31] and Mair et al. [32]. This implies that the effectiveness of probiotics is related to the specific growth period, probiotic strains, doses, etc., in piglets.

Intestinal morphology is an important indicator of intestinal health, usually evaluated according to the VH, CD, and VCR [33]. The intestine is the main site of the digestion and absorption of nutrients, and villous health is a key factor influencing nutrient absorption. Shortened villi and deeper crypts hinder nutrient absorption [34]. The results obtained here show that challenge with LPS increased the CD in the ileum and decreased the VCR in the ileum; LPS induced acute intestinal injury in the piglets. However, LAB supplementation reversed the structural damage to the intestinal barrier induced by LPS, in agreement with the previous finding that piglets given *Lactobacillus plantarum* had higher villous height and VCR and lower crypt depth [35]. Thus, dietary LAB supplementation may improve intestinal morphology in LPS-challenged piglets.

Since the intestine is a direct “channel” to the outside world, it is easily irritated by external factors, which may lead to the disruption of its barrier function. This dysfunction caused by oxidative stress is an important source of digestive tract diseases [36], commonly manifesting as increased intestinal permeability [37]. Numerous studies indicate that oxidative stress triggers a significant decrease in intestinal transepithelial electrical resistance and a significant increase in intestinal permeability [38,39,40]. The intestinal barrier is primarily regulated by tight junctions, which comprise a well-organized junctional structure located at the tip of the intestinal epithelium [41]. The tight junctions between intestinal epithelial cells are mainly composed of occludin, claudins, ZO-1, and junctional adhesion molecules, as well as cytoplasmic proteins [42]. Once the intestinal barrier is compromised, the intestinal epithelium releases DAO into the bloodstream [43], so serum DAO is regarded as a marker for intestinal permeability. As expected, the current results show that higher DAO activity in the serum of LPS-challenged piglets was related to the destruction of intestinal integrity and epithelial function. However, dietary LAB supplementation repaired the intestinal permeability. Abnormal distributions of tight junction proteins are significantly related to intestinal diseases and cause susceptibility to disruption by oxidative stress. A study in a piglet model of oxidative stress revealed that oxidative stress significantly downregulates tight junction protein expression and increases serum DAO activity, accompanied by altered intestinal morphology [44]. Therefore, the fact that the protein expression of ZO-1 (jejunum and ileum) and claudin-1 (ileum) was significantly decreased in the LPS group in the present study appears justifiable. LAB supplementation enhanced the expression of tight junction proteins, including occludin and claudin-1 in both the jejunum and ileum, and ZO-1 in the ileum, demonstrating that LAB may contribute to improving intestinal barrier function and alleviating tissue injury. Consistent with this, Sun et al. also found that the protein expression of occludin, claudin-1, and ZO-1 was significantly enhanced by the addition of *Lactobacillus salivarius* to the diet [26]. These results indicate that LAB may improve the integrity of the intestinal epithelial barrier.

Redox homeostasis is indispensable for human and animal cells, tissues, and organs [45] and mainly depends on the dynamic balance between the oxidation and antioxidant systems. When this balance is disrupted, oxidative stress occurs easily as a result of excessive reactive oxygen species production or the inactivation of the antioxidant system [46,47]. Oxidative stress oxidizes biological macromolecules (e.g., proteins, nucleic acids, and lipids), resulting in changes in structure and physiological function [47,48]. MDA, produced by lipid peroxidation, is often treated as a sign of oxidative injury, with 8-OHdG indicating DNA oxidation. In animals, the antioxidant defense system protects against oxidative damage and mainly comprises non-enzymatic antioxidants (such as GSH and GSSH) and antioxidant enzymes (such as CAT, SOD, GSH-Px, GR, and HO-1) [49]. In this study, LPS challenge decreased the activity of CAT and GSH-Px in the intestinal mucosae and serum, and GR and SOD in the serum, which was similar to the results of previous research [22]. These results fully illustrate the success of this study in modeling oxidative stress with the LPS challenge. Additionally, based on the decreased MDA contents in the serum and jejunal mucosa, the lipid peroxidation damage induced by LPS was shown to be relieved by LAB supplementation. Many reports have shown similar effects of *Lactobacillus* [26,50,51]. Moreover, the LPS-induced decrease in the activity of antioxidant enzymes (GR in the serum and ileum; CAT in the ileum; SOD in the jejunum) was reversed by LAB supplementation. *L. rhamnosus Gorbach-Goldin* alleviates alcohol-induced oxidative stress in the liver and protects it from oxidative damage [52]. A clinical trial in pregnant women showed that the antioxidant enzyme activity and GSH level in the serum were increased by *Lactobacillus* [53]. Interestingly, we found that the antioxidant capacity of jejunum and ileum was not identical, as evidenced by differences in SOD activity and 8-OHdG contents. This may be related to the development of different intestinal segments and the colonization site of LAB. Overall, LAB may protect against intestinal oxidative damage by enhancing antioxidant enzymes, maintaining redox homeostasis, but further studies are needed to confirm the potential mechanisms.

Numerous studies have reported that probiotics exert beneficial effects mainly through the recognition of the probiotics by TLRs, which activates corresponding signaling pathways and triggers appropriate responses [54,55]. The upregulation of the Nrf2 pathway, which protects against oxidative stress and injury, has been reported [56]. Nrf2 activates a series of downstream phase II detoxification and antioxidant enzymes upon exposure to oxidative stressors [57]. HO-1 is an antioxidant enzyme with antioxidant-based cytoprotective effects [58,59], whose gene expression is mediated through an Nrf2-activated redox-dependent signaling pathway, and it is involved in the activation of Btk [18,60]. TLR agonists may act as activators of the Nrf2 pathway, promoting the expression of antioxidant molecules [61]. According to Yin et al., inducers of TLR4, TLR2, and TLR3 may activate the Nrf2–ARE pathway and its downstream antioxidant enzymes (HO-1) in vivo and in vitro [62]. To further investigate the mechanism by which LAB alleviates LPS-induced oxidative stress in piglets, Western blotting was used to assay proteins related to the TLR–Btk–Nrf2 signaling pathway, showing that the protein expression of TLR2, TLR4, and Btk in the jejunum and Btk, HO-1, and Nrf2 in the ileum was decreased in LPS-challenged piglets, but that of TLR2 (jejunum), TLR4 (jejunum and ileum), Btk (jejunum and ileum), HO-1 (jejunum), and Nrf2 (jejunum and ileum) was increased by LAB. It can be assumed that LAB is able to trigger an antioxidant effect by activating TLR–Btk–Nrf2 signaling. In mice challenged with *Salmonella enterica*, the expression of TLR2, TLR4, and TLR9 in the small intestine was increased by *Lactobacillus* [55]. Xu et al. also found that *Lactobacilli* exerts an antioxidative effect by raising the Nrf2 protein level [63]. *Bacillus* was also found to activate the Nrf2 signaling pathway [64]. Collectively, these findings suggest that the activation of the TLR–Btk–Nrf2 signaling pathway by LAB may account for the latter’s antioxidant effects in the intestinal tract. Therefore, further studies on these antioxidant mechanisms are warranted.

## 5. Conclusions

In conclusion, the present results confirmed that LAB supplementation exhibit beneficial effects on improving intestinal integrity when piglets suffered LPS stress, which is supported by sound reasons: intestinal mucosa structure and barrier function are improved, what’s more, oxidative stress mediated by the TLRs-Btk-Nrf2 signaling pathway is alleviated. This study provides theoretical basis for applying probiotics for the treatment and prevention of intestinal diseases in piglets.

## Figures and Tables

**Figure 1 antioxidants-10-00468-f001:**
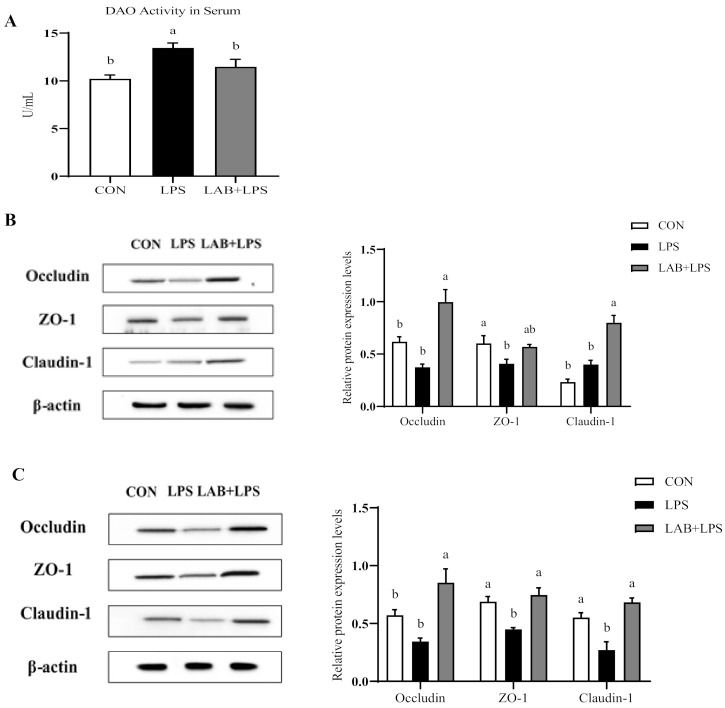
The effects of LAB supplementation on the serum diamine oxidase activity and intestinal tight junction protein expression in weaned piglets. (**A**) DAO activity in the serum. (**B**) The jejunal protein expression, based on Western blotting and statistical analysis, of occludin/β-actin, ZO-1/β-actin, and claudin-1/β-actin. (**C**) The ileal protein expression, based on Western blotting and statistical analysis, of occludin/β-actin, ZO-1/β-actin, and claudin-1/β-actin. Data are presented as means ± SEMs (*n* = 6). ^a,b^ Mean values within a row with different superscript letters indicate significant differences (*p* < 0.05). CON—non-challenged piglets fed a basal diet; LPS—LPS-challenged piglets fed a basal diet; LAB + LPS—LPS-challenged piglets fed a basal diet supplemented with 0.2% LAB. DAO—diamine oxidase; ZO-1—zonula occluden-1.

**Figure 2 antioxidants-10-00468-f002:**
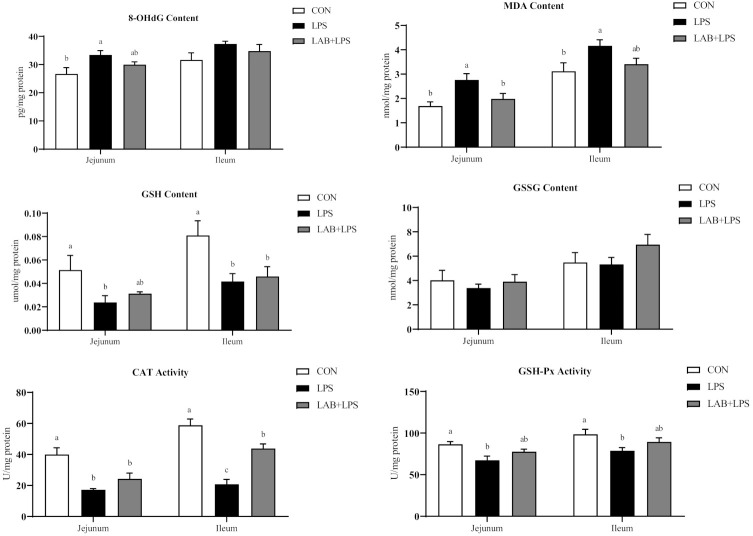
The effects of LAB supplementation on the intestinal mucosal oxidative statuses of weaned piglets. Data are presented as means ± SEM (*n* = 6). ^a,b,c^ Mean values within a row with different superscript letters indicate significant differences (*p* < 0.05). CON—non-challenged piglets fed a basal diet; LPS—LPS-challenged piglets fed a basal diet; LAB + LPS—LPS-challenged piglets fed a basal diet supplemented with 0.2% LAB. 8-OHdG—8-hydroxy-2-deoxyguanosine; MDA—malondialdehyde; GSH—glutathione; GSSG—oxidized glutathione; CAT—catalase; GSH-Px—glutathione peroxidase; GR—glutathione reductase; SOD—superoxide dismutase.

**Figure 3 antioxidants-10-00468-f003:**
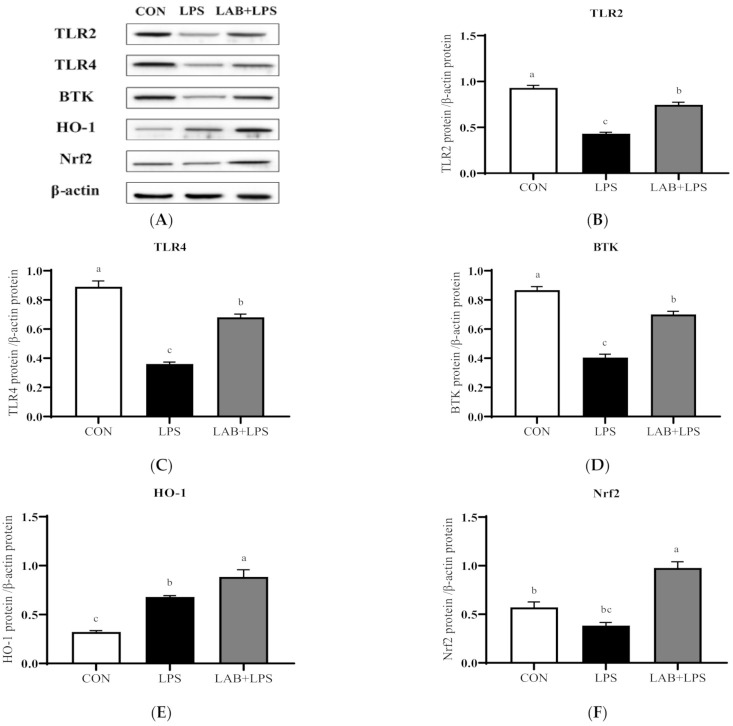
The effects of LAB supplementation on the protein expression of TLRs, Btk, and Nrf2 in the jejuna of weaned piglets. Expression based on Western blotting (**A**) and quantification for (**B**) TLR2, (**C**) TLR4, (**D**) Btk, (**E**) HO-1, and (**F**) Nrf2. Data are presented as means ± SEMs (*n* = 6). ^a,b,c^ Mean values within a row with different superscript letters indicate significant differences (*p* < 0.05). CON—non-challenged piglets fed a basal diet; LPS—LPS-challenged piglets fed a basal diet; LAB + LPS—LPS-challenged piglets fed a basal diet supplemented with 0.2% LAB. TLR2—Toll-like receptor 2; TLR4—Toll-like receptor 4; BTK—Bruton’s tyrosine kinase; HO-1—hemeoxygenase-1; Nrf2—nuclear factor erythroid 2-related factor 2.

**Figure 4 antioxidants-10-00468-f004:**
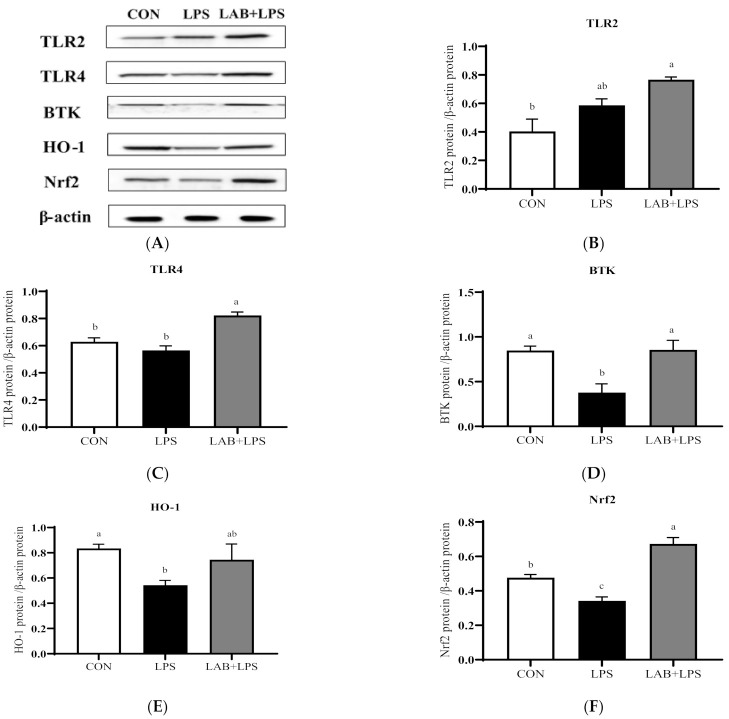
The effects of LAB supplementation on the protein expression of TLRs, Btk, and Nrf2 in the ilea of weaned piglets. Expression based on Western blotting (**A**) and quantification for (**B**) TLR2, (**C**) TLR4, (**D**) Btk, (**E**) HO-1, and (**F**) Nrf2. Data are presented as means ± SEM (*n* = 6). ^a,b,c^ Mean values within a row with different superscript letters indicate significant differences (*p* < 0.05). CON—non-challenged piglets fed a basal diet; LPS—LPS-challenged piglets fed a basal diet; LAB + LPS—LPS-challenged piglets fed a basal diet supplemented with 0.2% LAB. TLR2—Toll-like receptor 2; TLR4—Toll-like receptor 4; BTK—Bruton’s tyrosine kinase; HO-1—hemeoxygenase-1; Nrf2—nuclear factor erythroid 2-related factor 2.

**Table 1 antioxidants-10-00468-t001:** Ingredient composition of the basal diet (on an as-fed basis)**.**

Item	Content
Ingredient, %	
Extruded corn	50.00
Soybean meal, 43% crude protein	19.20
Extruded soybean	13.30
Fish meal	2.50
Whey powder	10.00
l-Lysine HCl, 78%	0.28
l-Methionine, 98%	0.21
l-Threonine, 98%	0.10
Dicalcium phosphate	0.75
Limestone	0.76
Vitamin and mineral Premix ^a^	2.00
Nutrient composition ^b^	
Digestible energy (MJ/kg)	14.57
Crude protein (%)	19.12
Calcium (%)	0.77
Available phosphorus (%)	0.45
Lysine (%)	1.38
Methionine (%)	0.52
Threonine (%)	0.85

^a^ Premix supplied per kg diet: retinyl acetate, 5512 IU; cholecalciferol, 2200 IU; dl-α-tocopheryl acetate, 30 IU; menadione sodium bisulfite complex, 4 mg; riboflavin, 5.22 mg; d-calcium–pantothenate, 20 mg; niacin, 26 mg; vitamin B 12, 0.01 mg; Mn (MnSO_4_·H_2_O), 63.6 mg; Fe (FeSO_4_·H_2_O), 90 mg; Zn (ZnSO_4_·7H_2_O), 103.5 mg; Cu (CuSO_4_·5H_2_O), 100 mg; I (CaI_2_), 0.2 mg; Se (Na_2_SeO_3_), 0.2 mg. ^b^ Calculated value from data provided by Feed Database in China.

**Table 2 antioxidants-10-00468-t002:** The effects of *Lactobacillus delbrueckii* (LAB) supplementation on the growth performance of weaned piglets.

Items	CON	LPS	LAB + LPS	*p*-Value
Initial BW (kg)	7.13 ± 0.45	7.35 ± 0.18	6.94 ± 0.08	0.59
Final BW (kg)	14.55 ± 0.65	13.81 ± 0.64	13.9 ± 0.54	0.66
ADFI (g)	549.50 ± 48.33	561.50 ± 49.26	553.50 ± 23.80	0.97
ADG (g)	265.00 ± 9.26	230.75 ± 21.44	248.75 ± 20.02	0.43
F/G	2.07 ± 0.13	2.50 ± 0.32	2.29 ± 0.28	0.52

Data are presented as means ± SEM (*n* = 6). CON—non-challenged piglets fed a basal diet; LPS—LPS-challenged piglets fed a basal diet; LAB + LPS—LPS-challenged piglets fed a basal diet supplemented with 0.2% LAB. BW—body weight; ADFI—average daily feed intake; ADG—average daily gain; F/G—feed/gain ratio.

**Table 3 antioxidants-10-00468-t003:** The effects of LAB supplementation on the intestinal morphology of weaned piglets.

Items	CON	LPS	LAB + LPS	*p*-Value
Jejunum				
Villus height (um)	269.40 ± 21.66	257.59 ± 15.36	253.85 ± 15.26	0.81
Crypt depth (mm)	72.15 ± 2.59 ^a,b^	78.18 ± 5.59 ^a^	61.36 ± 4.55 ^b^	0.05
VCR	3.92 ± 0.27	3.46 ± 0.36	4.1 ± 0.24	0.32
Ileum				
Villus height (um)	284.21 ± 14.22	263.5 ± 22.84	250.08 ± 10.30	0.37
Crypt depth (mm)	82.43 ± 3.22 ^b^	102.75 ± 6.62 ^a^	73.58 ± 3.27 ^b^	< 0.01
VCR	3.60 ± 0.32 ^a^	2.78 ± 0.24 ^b^	3.47 ± 0.17 ^a,b^	0.08

Data are presented as means ± SEM (*n* = 6). ^a,b^ Mean values within a row with different superscript letters were significantly different (*p* < 0.05). CON—non-challenged piglets fed a basal diet; LPS—LPS-challenged piglets fed a basal diet; LAB + LPS—LPS-challenged piglets fed a basal diet supplemented with 0.2% LAB. VCR—villus height-to-crypt depth ratio.

**Table 4 antioxidants-10-00468-t004:** The effects of LAB supplementation on the serum oxidative statuses of weaned piglets.

Items	CON	LPS	LAB + LPS	*p*-Value
8-OHdG (pg/mL)	28.11 ± 3.07	36.38 ± 3.35	33.06 ± 2	0.17
MDA (nmol/mL)	11.93 ± 0.51 ^b^	15.07 ± 0.59 ^a^	12.07 ± 0.94 ^b^	0.01
GSH (umol/mL)	0.2 ± 0.04	0.13 ± 0.03	0.16 ± 0.03	0.48
GSSG (nmol/mL)	17.61 ± 1.54	17.22 ± 1.06	17.88 ± 0.72	0.95
CAT (U/mL)	228.06 ± 20.76 ^a^	136.8 ± 7.49 ^b^	164.94 ± 15.07 ^b^	<0.01
GSH-Px (U/mL)	58.05 ± 4.68 ^a^	43.27 ± 4.09 ^b^	54.49 ± 3.63 ^a,b^	0.07
GR (U/mL)	149.7 ± 9.56 ^b^	115.49 ± 12.33 ^c^	206.46 ± 7.14 ^a^	<0.01
SOD (U/mL)	4.19 ± 0.46 ^a^	2.88 ± 0.3 ^b^	2.54 ± 0.09 ^b^	0.01

Data are presented as means ± SEM (*n* = 6). ^a,b,c^ Mean values within a row with different superscript letters indicate significant differences (*p* < 0.05). CON—non-challenged piglets fed a basal diet; LPS—LPS-challenged piglets fed a basal diet; LAB + LPS—LPS-challenged piglets fed a basal diet supplemented with 0.2% LAB. 8-OHdG—8-hydroxy-2-deoxyguanosine; MDA—malondialdehyde; GSH—glutathione; GSSG—oxidized glutathione; CAT—catalase; GSH-Px—glutathione peroxidase; GR—glutathione reductase; SOD—superoxide dismutase.

## Data Availability

The data used to support the findings of this study are available from the corresponding author upon request.

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
