# Peer review of "Lactobacillus delbrueckii Protected Intestinal Integrity, Alleviated Intestinal Oxidative Damage, and Activated Toll-Like Receptor–Bruton’s Tyrosine Kinase–Nuclear Factor Erythroid 2-Related Factor 2 Pathway in Weaned Piglets Challenged with Lipopolysaccharide"

_antioxidants, 2021, doi:10.3390/antiox10030468_

Round 1

Reviewer 1 Report

This is an interesting study reporting important findings. My major concern is that authors did not mention NF-kB and its possible involvement in protective effects of Lactobacillus. I believe that at lest in discussion it should be mentioned.

Author Response

Dear Reviewer:

Thank you for your letter and comments concerning our manuscript entitled “Lactobacillus delbrueckii Protected Intestinal Integrity, Allevi-ated Intestinal Oxidative Damage, and Activated Toll-like re-ceptor–Bruton's tyrosine kinase–nuclear factor erythroid 2-related factor 2 Pathway in Weaned Piglets Challenged with Lipopolysaccharide” (antioxidants-1113851). Those comments are all valuable and very helpful for revising and improving our paper, as well as the important guiding significance to our researches. We have studied comments carefully and have made a correction which we hope meet with approval. Revised portion are marked in red in the manuscript. The main corrections in the paper and the response to the reviewer’s comments are as following:

-Reviewer 1-

Comments to the Author

This is an interesting study reporting important findings. My major concern is that authors did not mention NF-kB and its possible involvement in protective effects of Lactobacillus. I believe that at lest in discussion it should be mentioned.

A: Thanks for reviewer’s suggestion. The nuclear factor kappa B (NFκB) family of transcription factors is a key regulator of immune development, immune responses, and inflammation. Numerous studies have shown that probiotics exert immunomodulatory functions in relation to the NF-KB signaling pathway. We have already focused on the effect of LAB on immune regulation in weaned piglets in another manuscript. In this manuscript, we focus on the mechanism study of LAB to alleviate oxidative damage in weaned piglets.

Reviewer 2 Report

Comments on Manuscript Antioxidants-113851

The aim of the present study was to investigate whether Lactobacillus delbrueckii (LAB) supplementation could restore intestinal integrity and alleviate oxidative stress/damages in lipopolysaccharide (LPS)-challenged piglets.

The obtained results indicated that LAB supplementation exhibited beneficial effects on improving intestinal mucosa structure and barrier function. Moreover, it was shown that oxidative stress mediated by the TLRs-Btk-Nrf2 signaling pathway was alleviated.

Nevertheless, this work provides theoretical basis for applying probiotics for the prevention of intestinal diseases in piglets.

General Impression:

The manuscript describes work which has been executed to a high standard and is interesting. Furthermore, the manuscript is well written, the introduction provides a detailed background to the project citing the work of others in the field. Redox homeostasis and intestinal barrier function disciplines are certainly interesting ones. The procedures and methods are pretty comprehensive and well documented. The statistical analysis of the results seems pretty thorough and well-presented. Overall, the results are clearly presented and discussed, make a significant contribution to knowledge in the field and are summarized in a concluding statement at the end.

Minor suggestions for improvement:

  • Please indicate city and country for all manufacturers/companies which provided materials.
  • A schematic interpretation (or graphical abstract) of the proposed TLR–Btk–Nrf2 signaling pathway triggered by LAB would be very nice.

Author Response

Dear Reviewer:

Thank you for your letter and comments concerning our manuscript entitled “Lactobacillus delbrueckii Protected Intestinal Integrity, Allevi-ated Intestinal Oxidative Damage, and Activated Toll-like re-ceptor–Bruton's tyrosine kinase–nuclear factor erythroid 2-related factor 2 Pathway in Weaned Piglets Challenged with Lipopolysaccharide” (antioxidants-1113851). Those comments are all valuable and very helpful for revising and improving our paper, as well as the important guiding significance to our researches. We have studied comments carefully and have made a correction which we hope meet with approval. Revised portion are marked in red in the manuscript. The main corrections in the paper and the response to the reviewer’s comments are as following:

Comments to the Author

The aim of the present study was to investigate whether Lactobacillus delbrueckii (LAB) supplementation could restore intestinal integrity and alleviate oxidative stress/damages in lipopolysaccharide (LPS)-challenged piglets. The obtained results indicated that LAB supplementation exhibited beneficial effects on improving intestinal mucosa structure and barrier function. Moreover, it was shown that oxidative stress mediated by the TLRs-Btk-Nrf2 signaling pathway was alleviated. Nevertheless, this work provides theoretical basis for applying probiotics for the prevention of intestinal diseases in piglets.

Minor suggestions for improvement:

Please indicate city and country for all manufacturers/companies which provided materials. A schematic interpretation (or graphical abstract) of the proposed TLR–Btk–Nrf2 signaling pathway triggered by LAB would be very nice.

A: Thanks for reviewer’s suggestion. The information of materials has been added. In addition, graphical abstract has been added.

Reviewer 3 Report

The manuscript is very interesting, However I can not see if

The experiment was approved by the Animal Experimentation Ethics Committee in order to approve
the protection of animals used for scientific purposes.

Please insert that informtaion into the manuscript. apporved number

Author Response

Dear Reviewers:

Thank you for your letter and comments concerning our manuscript entitled “Lactobacillus delbrueckii Protected Intestinal Integrity, Allevi-ated Intestinal Oxidative Damage, and Activated Toll-like re-ceptor–Bruton's tyrosine kinase–nuclear factor erythroid 2-related factor 2 Pathway in Weaned Piglets Challenged with Lipopolysaccharide” (antioxidants-1113851). Those comments are all valuable and very helpful for revising and improving our paper, as well as the important guiding significance to our researches. We have studied comments carefully and have made a correction which we hope meet with approval. Revised portion are marked in red in the manuscript. The main corrections in the paper and the response to the reviewer’s comments are as following:

Comments to the Author

The manuscript is very interesting, However I can not see if The experiment was approved by the Animal Experimentation Ethics Committee in order to approve the protection of animals used for scientific purposes. Please insert that informtaion into the manuscript. apporved number.

A: Thanks for reviewer’s suggestion, the apporved number has been added.

Reviewer 4 Report

The manuscript is interesting, but I am not sure if it fits well within the antioxidant journal as effects are not only of oxidative nature, they are as well inflammatory effects. There are some short comings in the manuscript which raised my concerns. See specific comments below.

Line

3-4        please do not use abbreviations in the title

18         please give the LAB dose in KBE and not in %

19         please include dose of LPS (? Kg/body weight)

47         reference No 10 cannot be used as reference here it is about natural antioxidants and prebiotics (not probiotics)as  ROS scavengers…please include a reference which supports your hypothesis number 11  looks fine

49         GPx and CAT…abbreviations should be introduced before first use

48/49    The change of the activity of the different antioxidant enzyme systems are not easy to interpretate as not only oxidative stress but as well inflammatory mediators may change their expression/ activity…positive interpretation  means an increased activity can be interpretated as a higher antioxidative capability …negative interpretation means oxidative stress or inflammatory process leads to an increase  in antioxidant enzyme activity. Therefore it is necessary that you explain what happens here, has LPS injection an inflammatory or oxidative effect…if it has both, the question arises which one is the most prominent and is for example the oxidative stress effect dependent on the inflammatory reaction or the other way around. . There is more information needed.

55         they may affect the redox state of the body but they don’t regulate it….please rewrite

68         Reference 20 by the same authors, the same parameters were examined partly with different analysis methods in along term trial .. as it gives partly already answers to the questions stated in the introduction why is it not referenced in the introduction and why is this challenge study give more information??.

74         the diet fed is not a typical weaner diet,. Weaner diet usually contain upto 20%  milk components…please explain how this may affect the results and why such a diet was chosen

78         This is not clear to me, as it looks the animals were fed the diets and on the first day after the experiment they were challenged with LPS and slaughtered after the experiment. However that means that all differences shown may have already been existent before LPS application.

             It would have been better if pigs without LPS challenge would have been slaughtered as control, so one could see the change within each group.

84         humanely killed is nonsense… please indicate how they were killed. Blood samples collected from where, how were they centrifuged (temperature g-force etc..). Please give precisely the site from which  the sample were collected. How were the samples harvested, was intestinal contents not removed?, flushed with buffered solution etc….This needs to be improved. Description of material and method must allow to repeat the experiment!!!

91         give supplier details for light microscopy

94         centrifuged …please include details for centrifugation

94         BCA ?? introduce abbreviations

127       serum diamine oxidase is not given in the table

141       this figure has no legend!!

155-159             what a mess…this is not acceptable as description please divide this into at least two sections. Firstly serum and then intestinal data (and then you might think about to split intestinal data into jejunum and ileum results as they are not always consistent.

223-224             intestinal cells do not get repaired they will be replaced by newly differentiated cells from the crypt. So lower crypt depth means that the intestinal cells have a higher longevity, they last longer, possibly due to reduction of pathogens in the intestine which reduce longevity of intestinal cells

             I miss an explanation why many of the parameters have been significant changed in the jejunum but not in the ileum for example SOD, GSSG.

Why was vitamin E, the most important antioxidant within the body, not measured???

Author Response

Dear Reviewers:

Thank you for your letter and comments concerning our manuscript entitled “Lactobacillus delbrueckii Protected Intestinal Integrity, Allevi-ated Intestinal Oxidative Damage, and Activated Toll-like re-ceptor–Bruton's tyrosine kinase–nuclear factor erythroid 2-related factor 2 Pathway in Weaned Piglets Challenged with Lipopolysaccharide” (antioxidants-1113851). Those comments are all valuable and very helpful for revising and improving our paper, as well as the important guiding significance to our researches. We have studied comments carefully and have made a correction which we hope meet with approval. Revised portion are marked in red in the manuscript. The main corrections in the paper and the response to the reviewer’s comments are as following:

Comments to the Author

The manuscript is interesting, but I am not sure if it fits well within the antioxidant journal as effects are not only of oxidative nature, they are as well inflammatory effects. There are some short comings in the manuscript which raised my concerns. See specific comments below.

L.3-4 please do not use abbreviations in the title.

A: Thanks for reviewer’s suggestion, the title has been revised.

L.18 please give the LAB dose in KBE and not in %.

A: Thanks for reviewer’s suggestion, the dose of LAB has been revised.

L.19 please include dose of LPS (? Kg/body weight).

A: Thanks for reviewer’s suggestion, the dose of LPS has been added.

L.47 reference No 10 cannot be used as reference here it is about natural antioxidants and prebiotics (not probiotics)as  ROS scavengers…please include a reference which supports your hypothesis number 11 looks fine.

A: Thanks for reviewer’s suggestion, reference 10 has been revised.

L.49 GPx and CAT…abbreviations should be introduced before first use.

A: Thanks for reviewer’s suggestion, here it has been revised.

L.48/49 The change of the activity of the different antioxidant enzyme systems are not easy to interpretate as not only oxidative stress but as well inflammatory mediators may change their expression/ activity…positive interpretation  means an increased activity can be interpretated as a higher antioxidative capability …negative interpretation means oxidative stress or inflammatory process leads to an increase  in antioxidant enzyme activity. Therefore it is necessary that you explain what happens here, has LPS injection an inflammatory or oxidative effect…if it has both, the question arises which one is the most prominent and is for example the oxidative stress effect dependent on the inflammatory reaction or the other way around. . There is more information needed.

A: Thanks for reviewer’s suggestion. In recent years, evidence has been obtained that chemical processes involving redox reactions triggering cellular oxidative stress play critical roles in the pathophysiology of inflammation [Liaudet, L., Vassalli, G., & Pacher, P. (2009). Role of peroxynitrite in the redox regulation of cell signal transduction pathways. Frontiers in bioscience (Landmark edition), 14, 4809–4814. https://doi.org/10.2741/3569. Nathan, C., & Cunningham-Bussel, A. (2013). Beyond oxidative stress: an immunologist’s guide to reactive oxygen species. Nature Reviews Immunology, 13(5), 349–361. https://doi.org/10.1038/nri3423]. Due to biomolecular damage exceeding any capacity of repair, oxidative stress may precipitate cellular death and extracellular matrix (ECM) breakdown. Necrotic cells and damaged ECM in turn release various intracellular and extracellular molecules, which act as ‘alarmins’ triggering inflammatory cascades through recognition by PRRs [Chan, J. K., Roth, J., Oppenheim, J. J., Tracey, K. J., Vogl, T., Feldmann, M., Horwood, N., & Nanchahal, J. (2012). Alarmins: awaiting a clinical response. The Journal of clinical investigation, 122(8), 2711–2719. https://doi.org/10.1172/JCI62423 ]. Furthermore, oxidative stress conditions may induce various modifications within lipids and proteins, generating the so-called oxidation-specific epitopes, which act as potent DAMPs able to trigger innate immune responses through binding to multiple PRRs. Thus, inflammatory response is inextricably linked to oxidative stress. Lipopolysaccharide (LPS) is a Gram-negative bacterial outer membrane component. LPS triggers the inflammatory response, leading to the release of large numbers of endogenous inflammatory mediators, including tumor necrosis factor (TNF-α), interleukins (IL-4, IL-10, IL-13, IL-1), chemokines, adhesion molecules, reactive oxygen species (ROS), and reactive nitrogen species (RNS) [Hewett, J. A., & Roth, R. A. (1993). Hepatic and extrahepatic pathobiology of bacterial lipopolysaccharides. Pharmacological reviews, 45(4), 382–411. Yoshino, S., Sasatomi, E., & Ohsawa, M. (2000). Bacterial lipopolysaccharide acts as an adjuvant to induce autoimmune arthritis in mice. Immunology, 99(4), 607–614. https://doi.org/10.1046/j.1365-2567.2000.00015.x ]. Therefore, lipopolysaccharide models are widely used as inflammatory response and oxidative stress in piglet research models. In the experiment in Ref. 12, growing–finishing pigs did not receive LPS injection. And the pigs were allotted to dietary treatments including a basal diet or the basal diet supplemented with either aureomycin or 10.2×1010 CFU/g Lact. fermentum diet.

L.55 they may affect the redox state of the body but they don’t regulate it….please rewrite.

A: Thanks for reviewer’s suggestion, here it has been revised.

L.68 Reference 20 by the same authors, the same parameters were examined partly with different analysis methods in along term trial .. as it gives partly already answers to the questions stated in the introduction why is it not referenced in the introduction and why is this challenge study give more information??.

A: Thanks for reviewer’s suggestion. The trial in reference 20, which was conducted by our laboratory in 2018, examined antioxidant enzyme activity and gene expression in the intestine and found that LAB was able to ameliorate LPS-induced oxidative damage in the intestine. However, an in-depth mechanistic study was not performed. By reviewing a large amount of literature, we found that the mechanism by which Lactobacillus exerts its antioxidant function may be related to the activation of TLR-Btk-Nrf2 signaling pathway. Therefore, in 2019, we designed this experiment to more systematically and deeply explore the mechanism of LAB alleviating LPS-induced intestinal oxidative damage in weaned piglets. In addition, reference 20 has been added to the introduction.

L.74 the diet fed is not a typical weaner diet,. Weaner diet usually contain upto 20%  milk components…please explain how this may affect the results and why such a diet was chosen.

A: Thanks for reviewer’s suggestion. The basal diet (Table 1) was formulated to meet the nutrient requirements recommended by the National Research Council (NRC 2012), which was consistent with the formulation in the experiment of reference 20. It included 10% whey powder.

Table 1. Ingredient composition of the basal diet (on an as-fed basis).

Item

Content

Ingredient, %

Extruded corn

50.00

Soybean meal, 43% crude protein

19.20

Extruded soybean

13.30

Fish meal

2.50

Whey powder

10.00

L -Lysine HCl, 98%

0.28

L -Methionine, 98%

0.21

L -Threonine, 98%

0.10

Dicalcium phosphate

0.75

Limestone

0.76

Vitamin and mineral Premix a

2.00

Nutrient composition b

Digestible energy (MJ/kg)

14.57

Crude protein (%)

19.12

Calcium (%)

0.77

Available phosphorus (%)

0.45

Lysine (%)

1.38

Methionine (%)

0.52

Threonine (%)

0.85

a Premix supplied per kg diet: retinyl acetate, 5512 IU; cholecalciferol, 2200 IU; DL-α-tocopheryl acetate, 30 IU; menadione sodium bisulfite complex, 4 mg; riboflavin, 5.22 mg; D-calcium–pantothenate, 20 mg; niacin, 26 mg; vitamin B 12, 0.01 mg; Mn (MnSO4·H2O), 63.6 mg; Fe (FeSO4·H2O), 90 mg; Zn (ZnSO4·7H2O), 103.5 mg; Cu (CuSO4·5H2O), 100 mg; I (CaI2), 0.2 mg; Se (Na2SeO3), 0.2 mg. b Calculated value from data provided by Feed Database in China.

L.78 This is not clear to me, as it looks the animals were fed the diets and on the first day after the experiment they were challenged with LPS and slaughtered after the experiment. However that means that all differences shown may have already been existent before LPS application.

A: Thanks for reviewer’s suggestion. I apologize probably because I did not state the exact time in the abstract, which has now been revised. The experimental period is 28 days. At eight o'clock in the morning on Day 29, the challenged groups were injected with LPS from Escherichia coli (Escherichia coli O55:B5, Sigma Chemical, St. Louis, MO, USA) at a dose of 100 ug/kg body weight (BW); the CON group was injected with saline.

It would have been better if pigs without LPS challenge would have been slaughtered as control, so one could see the change within each group.

A: Thanks for reviewer’s suggestion. Similar to the experimental grouping of Song et al,. [ Song, Z.; Tong, G.; Xiao, K.; Jiao, L.; Ke, Y.; Hu, C. L-Cysteine protects intestinal integrity, attenuates intestinal inflamma-tion and oxidant stress, and modulates NF-κB and Nrf2 pathways in weaned piglets after LPS challenge. Innate Immunity. 2016, 22, 152-161. DOI: 10.1177/1753425916632303 ] and Yi et al,. [Yi, D., Hou, Y., Wang, L., Ding, B., Yang, Z., Li, J., … Wu, G. (2013). Dietary N-acetylcysteine supplementation alleviates liver injury in lipopolysaccharide-challenged piglets. British Journal of Nutrition, 111(1), 46–54. https://doi.org/10.1017/s0007114513002171 ], ‌ this experiment statistical significance was analyzed via one-way analysis of variance (ANOVA), followed by Duncan’s multiple range test (SPSS 26.0 software, Chicago, IL, USA). In addition, piglets in the control group were fed a basal diet and received 0.9% NaCl solution injection. The LPS group is an oxidative stress model group, which fed a basal diet and received LPS injection. The 0.2% LAB + LPS group were fed a basal diet supplemented with 0.2% LAB (2.01×1010 CFU/g) and received LPS injection.

L.84 humanely killed is nonsense… please indicate how they were killed. Blood samples collected from where, how were they centrifuged (temperature g-force etc..). Please give precisely the site from which  the sample were collected. How were the samples harvested, was intestinal contents not removed?, flushed with buffered solution etc….This needs to be improved. Description of material and method must allow to repeat the experiment!!!

A: Thanks for reviewer’s suggestion, the sample collection method has been revised.

L.91 give supplier details for light microscopy.

A: Thanks for reviewer’s suggestion, the information of the microscope has been revised.

L.94 centrifuged …please include details for centrifugation.

A: Thanks for reviewer’s suggestion, details for centrifugation has been described.

L.94 BCA ?? introduce abbreviations.

A: Thanks for reviewer’s suggestion, the abbreviation for BCA has been described.

L.127 serum diamine oxidase is not given in the table.

A: Thanks for reviewer’s suggestion, here it has been revised.

L.141 this figure has no legend!!

A: Thanks for reviewer’s suggestion, here it has been revised.

L.155-159 what a mess…this is not acceptable as description please divide this into at least two sections. Firstly serum and then intestinal data (and then you might think about to split intestinal data into jejunum and ileum results as they are not always consistent.

A: Thanks for reviewer’s suggestion, the results of this section have been revised.

L.223-224 intestinal cells do not get repaired they will be replaced by newly differentiated cells from the crypt. So lower crypt depth means that the intestinal cells have a higher longevity, they last longer, possibly due to reduction of pathogens in the intestine which reduce longevity of intestinal cells.

A: Thanks for reviewer’s suggestion, here it has been revised.

I miss an explanation why many of the parameters have been significant changed in the jejunum but not in the ileum for example SOD, GSSG.

A: Thanks for reviewer’s suggestion, here it has been revised.

Why was vitamin E, the most important antioxidant within the body, not measured???

A: Thanks for reviewer’s suggestion. The measurement of antioxidant indices in this experiment was similar to the study of Jiang et al [Jiang, J., Qi, L., Lv, Z., Jin, S., Wei, X., & Shi, F. (2019). Dietary stevioside supplementation alleviates lipopolysaccharide-induced intestinal mucosal damage through anti-inflammatory and antioxidant effects in broiler chickens. Antioxidants, 8(12), 575.] and Yi et al [Yi, D., Hou, Y., Wang, L., Ding, B., Yang, Z., Li, J., … Wu, G. (2013). Dietary N-acetylcysteine supplementation alleviates liver injury in lipopolysaccharide-challenged piglets. British Journal of Nutrition, 111(1), 46–54. https://doi.org/10.1017/s0007114513002171 ]. Vitamin E is a fat-soluble antioxidant that can protect the polyunsaturated fatty acids (PUFAs) in the membrane from oxidation, affect the production of reactive oxygen species (ROS) and reactive nitrogen species (RNS). Thank you very much for your suggestion, it is true that we did not consider it comprehensively, and we will consider measuring vitamin E content in future experiments.

Round 2

Reviewer 3 Report

Than you for your review. Authors revised all the comments.

Please rewrite approved by approved:

All the procedures were approved by the animal welfare committee of Hunan Agricultural University (apporved 73 number: ACC2019016).

Author Response

Dear Reviewer:

Thank you for your letter and comments concerning our manuscript entitled “Lactobacillus delbrueckii Protected Intestinal Integrity, Alleviated Intestinal Oxidative Damage, and Activated Toll-like receptor–Bruton's tyrosine kinase–nuclear factor erythroid 2-related factor 2 Pathway in Weaned Piglets Challenged with Lipopolysaccharide” (antioxidants-1113851). Those comments are all valuable and very helpful for revising and improving our paper, as well as the important guiding significance to our researchers. We have studied comments carefully and have made a correction which we hope meet with approval. Revised portion are marked in red in the manuscript. The main corrections in the paper and the response to the reviewer’s comments are as following:

Comments to the Author

Please rewrite approved by approved: All the procedures were approved by the animal welfare committee of Hunan Agricultural University (apporved number: ACC2019016).

A: Thanks for reviewer’s suggestion, the sentence has been rewritten as follows: All the procedures were approved by the Animal Welfare Committee of Hunan Agricultural University, Changsha, China (ACC2019016).

We appreciate for Editors/Reviewers’ warm work earnestly, and hope that the correction will meet with approval.

Once again, thank you very much for your good comments and suggestions.

Best regards.

Yours,

Fengming Chen

Reviewer 4 Report

Line 59 TLR–Btk (Bruton's tyrosine kinase)…should be changed into… Bruton's tyrosine kinase (TLR-Btk)

The answer for my comment Line48/49 should be shared with the reader and not only with the reviewer, the authors give a long abstract on inflammatory process and oxidative stress effect of LPS…it would have been much better if the authors would integrate this in the text

The comment on the diet composition should be as well shared with the reader and not only with the reviewer please  include full dietary composition in the manuscript.

Author Response

Dear Reviewer:

Thank you for your letter and comments concerning our manuscript entitled “Lactobacillus delbrueckii Protected Intestinal Integrity, Alleviated Intestinal Oxidative Damage, and Activated Toll-like receptor–Bruton's tyrosine kinase–nuclear factor erythroid 2-related factor 2 Pathway in Weaned Piglets Challenged with Lipopolysaccharide” (antioxidants-1113851). Those comments are all valuable and very helpful for revising and improving our paper, as well as the important guiding significance to our researchers. We have studied comments carefully and have made a correction which we hope meet with approval. Revised portion are marked in red in the manuscript. The main corrections in the paper and the response to the reviewer’s comments are as following:

Comments to the Author

Line 59 TLR–Btk (Bruton's tyrosine kinase)…should be changed into… Bruton's tyrosine kinase (TLR-Btk)

A: Thanks for reviewer’s suggestion, here it has been revised.

The answer for my comment Line48/49 should be shared with the reader and not only with the reviewer, the authors give a long abstract on inflammatory process and oxidative stress effect of LPS…it would have been much better if the authors would integrate this in the text.

A: Thanks for reviewer’s suggestion, these have been integrated in the discussion section.

The comment on the diet composition should be as well shared with the reader and not only with the reviewer please include full dietary composition in the manuscript.

A: Thanks for reviewer’s suggestion. Ingredient composition of the basal diet has been added.

We appreciate for Editors/Reviewers’ warm work earnestly, and hope that the correction will meet with approval.

Once again, thank you very much for your good comments and suggestions.

Best regards.

Yours,

Fengming Chen